## PERSPECTIVE

### A sound vision: *MYO7A* gene therapy reaches the inner ear

Shahar Taiber[1,2] (ID)
and Karen B. Avraham[1] (ID)

[1]*Department of Human Molecular Genetics and Biochemistry, Gray Faculty of Medical and Health Sciences and Sagol School of Neuroscience, Tel Aviv University, Tel Aviv, Israel*

[2]*Department of Otolaryngology/Head, Neck and Maxillofacial Surgery, Tel Aviv Sourasky Medical Center, Tel Aviv, Israel*

Email: karena@tauex.tau.ac.il

Handling Editors: Vaughan Macefield & Conny Kopp-Scheinpflug

The peer review history is available in the Supporting Information section of this article (https://doi.org/10.1113/JP290105#support-information-section).

The field of inner ear gene therapy has reached an exciting turning point. With clinical trials for otoferlin (*OTOF*) gene therapy currently underway across the US, China, the UK and France, we are witnessing the translation of decades of preclinical research into potential treatments for hereditary hearing loss. The announcement of these trials has been compared to the wait for the Rolling Stones to take the stage (Brigande 2024). Beyond *OTOF*, gene therapy approaches have been tested for more than twenty genes in mouse models of hearing loss, predominantly using recombinant adeno-associated virus (AAV) vectors. The work by Amariutei et al. (2025) represents a significant milestone in this context, successfully expanding this therapeutic repertoire by treating myosin VIIa (*MYO7A*)-related hearing impairment in the Shaker-1 (*Myo7a*$^{sh1/sh1}$) mouse model.

*MYO7A* holds the distinction of being the first gene discovered to cause hereditary hearing loss in humans (Weil et al., 1995). Remarkably, pathogenic variants in this gene are responsible for both a syndromic form of hearing loss, Usher syndrome type 1B (USH1B), and recessive and dominant non-syndromic hearing loss. *MYO7A* variants are clinically relevant because a recent study demonstrated that *MYO7A* contributes to 1.4% of all deafness cases, at least in Japan (Watanabe et al. 2024).

In USH1B, *MYO7A* pathogenic variants cause not only profound hearing loss, but also progressive vision loss and balance dysfunction. *MYO7A* encodes myosin VIIa, a motor protein found in the stereocilia of inner ear hair cells that is crucial for both the development and maintenance of the inner ear hair cell bundles.

The auditory phenotype of *MYO7A* variants varies depending on the mutation's nature and impact on protein stability. In *Myo7a*$^{sh1/sh1}$ mice, a point mutation at a poorly conserved site leads to normal hair bundle development, followed by subsequent degeneration and early-onset hearing loss; this mutation is analogous to the variants in USH1B. By contrast, mutations causing complete or near-complete absence of *Myo7a*, such as *Myo7a*$^{4626SB/4626SB}$, result in early defective hair bundle development and profound deafness. Earlier attempts to rescue hearing in the *Myo7a*$^{4626SB/4626SB}$ mice failed to produce detectable hearing improvement (Lau et al. 2023).

The study by Amariutei et al. (2025) marks the first successful rescue of hearing in a mouse model of congenital *MYO7A* hearing loss, representing a crucial step forward in addressing DFNA11, DFNB2 and the auditory component of USH1B. Amariutei et al. (2025) employed a dual AAV vector approach, used previously for *Myo7a* and other large genes, because the coding sequence is too large for a single AAV vector. The cDNA sequence was split across two AAV backbones, with splice-donor and splice-acceptor sequences incorporated at the 3′ and 5′ ends, respectively, to facilitate splicing-mediated reconstruction of full-length mRNA *in vivo*. The researchers tested both AAV8 and AAV9-PhP.eB capsids for delivery, finding similar transduction efficiency in inner ear hair cells.

The treatment demonstrated several key improvements, including structural and functional recovery of hair cells, as well as a modest improvement in hearing. Scanning electron microscopy revealed restoration of hair bundle structure, with treated mice showing improved morphology, characterized by increased numbers of inner hair cells with multiple rows of stereocilia. The study demonstrated rescue of mechanotransduction currents, albeit at reduced amplitude compared to wild-type

controls, and restoration of resting opening probability of mechanotransduction channels, indicating functional recovery at the hair cell level. Finally, auditory brainstem response testing showed a modest but significant 20–30 dB improvement compared to untreated mice. The extent of rescue could probably be enhanced through future innovations in gene delivery, as demonstrated with other deafness genes such as *TMC1*.

The failure of previous attempts to rescue hearing in nonsense-mutation models, combined with limited success in treating mutations affecting early stereocilia development, highlights the narrow therapeutic window for intervention. Once extensive hair cell structural damage occurs, recovery potential becomes severely restricted. In humans, the auditory system becomes functional during embryonic development, with evidence suggesting fetal hearing begins during the second trimester of pregnancy. This timeline suggests that early post-natal intervention may be insufficient for humans with *MYO7A* mutations, potentially necessitating *in utero* interventions despite their inherent complexities and ethical considerations.

The inner ear encompasses not only the cochlea, responsible for hearing, but also vestibular organs that detect linear and angular head accelerations, which are crucial for balance. Myosin VIIa is important for vestibular hair cell development, and Shaker-1 mice exhibit abnormal balance and circling behaviour. Future studies should assess whether gene therapy rescues vestibular function, in addition to hearing. Additionally, because some deafness genes affect central auditory system components, such as the auditory cortex, comprehensive evaluation through behavioral assays will be essential to determine whether the observed physiological recovery translates to meaningful hearing improvement.

This work represents a significant advancement in *Myo7a* gene therapy, successfully demonstrating hearing rescue in a challenging mouse model. Although therapeutic outcomes show promise, optimizing delivery methods and addressing the critical timing of intervention remain key challenges for clinical translation.

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

## Additional information

### Competing interests

No competing interests declared.

### Author contributions

S.T. and K.B.A. wrote the perspective.

### Funding

Breakthrough Award, 2711/22 of the Israel Science Foundation: Karen B. Avraham. ORION Program for Physician-Scientists, Tel Aviv Sourasky Medical Center: Shahar Taiber.

### Keywords

adeno-associated virus, deafness, gene therapy, hearing loss, usher syndrome

### Supporting information

Additional supporting information can be found online in the Supporting Information section at the end of the HTML view of the article. Supporting information files available:

**Peer Review History**

