## [Peer Review History · The Journal of Physiology]

A sound vision: *MYO7A* gene therapy reaches the inner ear

Karen B. Avraham and Shahar Taiber
DOI: 10.1113/JP290105

Corresponding author(s): Karen Avraham (karena@tauex.tau.ac.il)

The following individual(s) involved in review of this submission have agreed to reveal their identity: Walter Marcotti (Referee #1)

Review Timeline:

Submission Date: 22-Sep-2025

Accepted: 25-Sep-2025

Senior Editor: Vaughan Macefield

Reviewing Editor: Conny Kopp-Scheinflug

Transaction Report:

Dear Professor Avraham,

Re: JP-P-2025-290105 "**A sound vision: *MYO7A* gene therapy reaches the inner ear**" by Karen B. Avraham and Shahar Taiber

We are pleased to tell you that your paper has been accepted for publication in The Journal of Physiology.

Yours sincerely,

Vaughan Macefield
Senior Editor
The Journal of Physiology

If you would like to receive our 'Research Roundup', a monthly newsletter highlighting the cutting-edge research published in The Physiological Society's family of journals (The Journal of Physiology, Experimental Physiology, Physiological Reports, The Journal of Nutritional Physiology, and The Journal of Precision Medicine: Health and Disease), please click this link, fill in your name and email address and select 'Research Roundup':

<https://www.physoc.org/journals-and-media/membernews>

- You can help your research get the attention it deserves! Check out Wiley's free Promotion Guide for best-practice recommendations for promoting your work at: www.wileyauthors.com/eeo/guide. You can learn more about Wiley Editing Services which offers professional video, design, and writing services to create shareable video abstracts, infographics, conference posters, lay summaries, and research news stories for your research at: www.wileyauthors.com/eeo/promotion.

The Corresponding Author will receive an email from Wiley with details on how to register or log-in to Wiley Authors Services where you will be able to place an order

EDITOR COMMENTS

Reviewing Editor:

Thank you for your successful perspective. It will help the field to gain more visibility even beyond auditory research.

Senior Editor:

Thank you for your interesting Perspectives article, which had been reviewed favourably by the author of the original article. I am pleased to report that your article is considered acceptable for publication in The Journal of Physiology.

REFEREE COMMENTS

Referee #1:

Thank you so much for writing the Perspective on this paper.

This article provides a well-written and comprehensive summary of the field's current state. This is not surprising considering that Professor Avraham is a pioneer in discovering deafness genes (such as MYO6) and developing gene therapies.

The introductory paragraph effectively outlines the impact of AAV-based gene therapy in the field. This is followed by key facts linking MYO7A to hearing loss in humans and the phenotypic effects of Myo7a mutations in mice. The concluding paragraphs focus on our paper's primary findings and the limitations we face in developing a fully successful therapy.